# The Potential of Hsp90 in Targeting Pathological Pathways in Cardiac Diseases

**DOI:** 10.3390/jpm11121373

**Published:** 2021-12-16

**Authors:** Richard J. Roberts, Logan Hallee, Chi Keung Lam

**Affiliations:** 1Department of Biological Sciences, University of Delaware, Newark, DE 19716, USA; rjrobert@udel.edu; 2Department of Mathematical Sciences, University of Delaware, Newark, DE 19716, USA; lhallee@udel.edu

**Keywords:** Hsp90, fibrosis, hypertrophy, cardiomyopathy, heart failure, HF, signal transduction

## Abstract

Heat shock protein 90 (Hsp90) is a molecular chaperone that interacts with up to 10% of the proteome. The extensive involvement in protein folding and regulation of protein stability within cells makes Hsp90 an attractive therapeutic target to correct multiple dysfunctions. Many of the clients of Hsp90 are found in pathways known to be pathogenic in the heart, ranging from transforming growth factor β (TGF-β) and mitogen activated kinase (MAPK) signaling to tumor necrosis factor α (TNFα), G_*s*_ and G_*q*_ g-protein coupled receptor (GPCR) and calcium (Ca^2+^) signaling. These pathways can therefore be targeted through modulation of Hsp90 activity. The activity of Hsp90 can be targeted through small-molecule inhibition. Small-molecule inhibitors of Hsp90 have been found to be cardiotoxic in some cases however. In this regard, specific targeting of Hsp90 by modulation of post-translational modifications (PTMs) emerges as an attractive strategy. In this review, we aim to address how Hsp90 functions, where Hsp90 interacts within pathological pathways, and current knowledge of small molecules and PTMs known to modulate Hsp90 activity and their potential as therapeutics in cardiac diseases.

## 1. Introduction

### 1.1. Hsp90 as a Chaperone

The primary roles of chaperones in the cell are to help stabilize proteins during folding, assisting them to reach their active conformation, and regulate their degradation. Many proteins require chaperone activity to assume their active conformation, with 20–30% of mammalian proteins lacking native three-dimensional structure [1]. Chaperones are also critical in the heat shock response by preventing protein unfolding and misfolding due to environmental stressors, intracellular stressors, and mutations [2]. This stabilization allows cells to continue functioning in these suboptimal conditions. Heat shock proteins (Hsps) function to prevent protein aggregation by dissembling and refolding aggregates, labeling peptides for proteasomal degradation, and sequestering proteins via the spatial protein quality control mechanism [3,4]. This balance between protein stabilization and degradation is called proteostasis (protein homeostasis) and is vital to cell survival [1].

There are five major categories of heat shock proteins: small heat shock proteins (sHsps), Hsp60, Hsp70, Hsp90, and Hsp100. Each class has multiple isoforms with their own function [2]. This review is focused on Hsp90. The name heat shock protein 90 refers to the role it plays in the heat shock response as well as its molecular weight (90 kDa) which distinguishes it from other heat shock proteins. Hsp90 is highly conserved across many species ranging from *E. coli* to humans [5]. The Hsp90 chaperone family contains four isoforms in mammalian cells. These isoforms are Hsp90α, Hsp90β, glucose response protein 94 (Grp94), and tumor necrosis factor type 1 receptor-associated protein (TRAP1) [6]. Hsp90α/β operates in the cytosol while Grp94 localizes to the endoplasmic reticulum and TRAP1 to the inner mitochondrial space [7]. Hsp90α expression is inducible and regulated by heat shock factor 1 (HSF1), while Hsp90β is constitutively expressed [8]. Together, these two isoforms make up 1–2% of cytosolic proteins in normal homeostasis, and up to 4–6% when a cell is stressed [9]. It has been suggested that Hsp90α/β potentially interact with 10% of all cytosolic proteins [10] which demonstrates how important it is to understand the roles they play. Hsp90α/β have the same general function and interact with the same cochaperones across different cell types. The major difference between the two is how much Hsp90α is upregulated following heat shock relative to Hsp90β [8,11]. In this review, we will focus on the cytosolic Hsp90 (Hsp90α/β).

### 1.2. Hsp90 Structure

Hsp90 is expressed as a monomer, however, the homodimerization of these monomers is required for chaperone activity [12]. There has also been evidence of Hsp90α/β heterodimers in HEK293 cells, however, evidence for these heterodimers is not abundant [13]. The Hsp90 monomer contains four major domains that are critical in its function: n-terminal (NTD), charged linker (CL), middle (MD), and c-terminal (CTD) domains. The NTD is responsible for the ATPase activity which drives the conformation cycle of the enzyme. There is also a small part of the NTD that is referred to as the “lid” which closes ATP into the active site [14]. The MD interacts with substrate (or client) proteins, acts as a binding site for co-chaperones, and is involved in ATP hydrolysis. Upon ATP binding, the MDs undergo a dramatic shift in position and eventually cross over each other [14]. Connecting the MD and NTD is a charged linker (CL) which contributes flexibility during conformational shifts. The CL also seems to play a role in the regulation of Hsp90 conformation and chaperone cycle [15,16,17]. The CTD is largely involved in the dimerization of Hsp90 monomers to form the functional Hsp90 enzyme. Much like the “lid” in the NTD, the CTD contains a motif called MEEVD, which is derived from the single letter amino acid code. The MEEVD motif is important in many co-chaperone interactions. These cochaperones contain tetratricopeptide repeat (TPR) domains which facilitate the binding to MEEVD [18]. The various co-chaperone interactions play a huge role in driving Hsp90 function and ATPase activity.

### 1.3. Hsp90 Chaperone Cycle and Function

Hsp90 function is best represented as a cycle involving various co-chaperones that facilitate conformational changes (see Figure 1). The cycle begins with an Hsp90 dimer in an open conformation. Here, the middle domains are split far apart and the ATPase catalytic site in the NTDs are empty. The open conformation is stabilized by CDC37 (cell division cycle 37), HOP (Hsp70-Hsp90 organizing protein), and PPIase (peptidyl-prolyl cis-trans isomerase). CDC37 binds to the NTD of HSP90 and inhibits ATPase activity of the homodimer. The main function of CDC37 is the activation of kinase clients [19,20,21,22]. HOP inhibits ATPase activity and aids in the recruitment of various client proteins via Hsp70 recruitment [23,24]. There is also evidence that HOP interacts with components of the proteasome which may contribute to HSP90 client degradation [25]. Lastly, PPIases help fold client proteins via their activity [18,26]. PPIases commonly associated with Hsp90 are FKBP51, FKBP52, and CYP40 (FK506-binding proteins 51, 52, and peptidyl-prolyl cis-trans isomerase 40) [14].

While these co-chaperones are bound in the open state, ATP enters the ATPase site of each Hsp90 monomer causing the NTDs and MDs to begin moving toward each other. This is referred to as the intermediate state. The intermediate state transitions to closed state 1 once the NTDs meet and dimerize. During closed state 1, Aha1 (activator of Hsp90 ATPase activity) binds and stabilizes Hsp90 at the NTD and MD [27]. Upon binding, Aha1 activates the ATPase activity of Hsp90 by displacing HOP [28]. This activation drives the Hsp90 MDs to cross over each other, leading to closed state 2 [29]. In closed state 2, Aha1 is displaced by p23 (prostaglandin E synthase 3) which stabilizes the closed state 2 and inhibits ATPase activity. It is proposed that the inhibition of ATPase activity by p23 allows Hsp90 to maintain its closed state 2 conformation which has a high affinity for client proteins [30]. Closed state 2 has been recognized as an important step in client protein maturation due to its increased client affinity [30,31]. There is also evidence of FKBP51 binding along with p23 at this stage suggesting these complexes may be specific to FKBP51-recruited clients [32]. ATP is hydrolyzed during closed state 2. Following ATP hydrolysis, the co-chaperones dissociate from the complex, ADP and Pi are released, and the closed state 2 conformation transitions back to the open state. The Hsp90 dimer is then able to repeat the cycle.

## 2. Hsp90 in Cardiomyopathy

Cardiomyopathy refers to electrical or muscular dysfunction in heart tissue, which can be induced by a diverse set of pathological conditions or genetic factors [33]. Cardiomyopathies come in five general classifications: ischemic, dilated (DCM), hypertrophic (HCM), arrhythmogenic (ACM), and restrictive (RCM) [34]. Ischemic cardiomyopathy occurs when the heart muscle is damaged from a lack of oxygen typically from coronary artery disease and atherosclerosis [35]. DCM, HCM, ACM, and RCM are typically caused by genetic factors affecting the myocardium, which can be further exacerbated by pathophysiological conditions like hypertension [34]. Fibrotic and hypertrophic signaling in these conditions becomes imbalanced leading to their development and advancement [36,37]. Uncontrolled cardiomyopathy will ultimately lead to congestive heart failure (HF). The clinical syndrome of HF places a considerable burden on the United States healthcare system. Estimates pin the total cost of HF to increase to 70 billion dollars annually by 2030 [38]. In this regard, Hsp90 plays an important role in many of these cardiomyopathy-related pathways. Our current understanding of the Hsp90 interactome highlights the potential for targeting Hsp90 in the prevention of fibrosis, hypertrophy, and cell death response, which are crucial contributors of cardiomyopathy development (see Figure 2) [22].

### 2.1. Pathways Regulated by Hsp90 in the Heart

#### 2.1.1. TGF-β Signaling

Transforming growth factor β (TGF-β) is a potent cytokine which plays an important role in cellular responses, such as angiogenesis, fibrosis, and immune response [39]. Induction of higher extracellular TGF-β levels can occur via mechanical overload typically in the form of hypertension, myocardial infarction, as well as ischemia/reperfusion (IR) injury [40]. TGF-β has been shown to induce cardiac hypertrophy as well as cardiac fibrosis [41]. These two responses occur from the difference between canonical and non-canonical TGF-β signaling. In the canonical cascade, the receptor is activated via autophosphorylation upon ligand binding. Another protein called activin receptor-like kinase 1 (ALK1) dimerizes with the receptor and is also phosphorylated. ALK1 is a kinase which aids in the phosphorylation of Smad1 and Smad5 proteins. The Smad1/5 complex joins with Smad4. This trimer is then transported to the nucleus where it acts as a transcription factor, activating genes involved in the fibrotic response and extracellular matrix (ECM) production [42]. Hsp90 has been shown to stabilize Smads and potentially aid in their translocation to the nucleus [43]. It has also been implicated in the stabilization of the TGF-β receptor which prevents the degradation of the receptor via SMURF-mediated ubiquitination [44,45].

The non-canonical TGF-β signaling is mediated by ALK5. The ALK5/TGF-βR dimer can activate Smad2 and Smad3 which form a trimer with Smad4 which is translocated to the nucleus similarly as the canonical pathway with Smads 1 and 5. Interestingly, non-canonical non-smad signaling can also cross-talk with a few other signaling cascades. These are proteins such as rat sarcoma virus (Ras) and TAK1 which are known Hsp90 clients and are vital in leading to transcriptional changes [46,47,48]. Ras phosphorylates ERK which acts as a transcription factor, TAK1 phosphorylates p38 which goes on to activate a wide range of transcription factors, some of which are implicated in cardiac hypertrophy [41,49,50]. Non-Smad TGF-β signaling can also stimulate phosphatidylinositol 3-kinase (PI3K) signaling which activates mammalian target of rapamycin (mTOR) [51]. All of these proteins and their relation to Hsp90 are covered in the MAPK Signaling and PI3K Signaling sections.

#### 2.1.2. MAPK Signaling

MAPK signaling is responsible for expression of proteins involved in cell proliferation, differentiation, development, apoptosis, and inflammation [52]. In the heart, MAPK signaling is induced by growth factors [53]. The varying responses depend on which arm of the signaling cascade is activated. As mentioned before, TGF-β is able to activate the p38 pathway of MAPK which goes on to express proteins involved in all categories previously listed. Both p38 and the kinase which activates it, mitogen-activated protein kinase kinase kinase 7 (MAP3K7 or TAK1), have been found to be Hsp90 clients [46,47]. Another part of MAPK signaling relevant in heart tissue is extracellular-signal-regulated kinase 1 and 2 (ERK1/2), Here, signaling is activated in the well-known MAPK cascade Ras-Raf-Mek-Erk typically through activation of a tyrosine kinase receptor (RTK) [54]. This pathway is known to upregulate proliferative genes as well as those involved in differentiation and development [52]. Within this pathway, Hsp90 has been shown to chaperone for MEK1, A-Raf, B-Raf, Raf-1, ERK, p90RSK, STAT3, and STAT5 [55,56]. The JNK arm of MAPK signaling is activated by a different cascade. A few different kinases (including TAK1) can phosphorylate MKK4 and MKK7 which can phosphorylate c-Jun N-terminal kinases (JNK). JNK goes on to activate many transcription factors responsible for regulating genes related to proliferation, differentiation, and apoptosis [52]. Again, Hsp90 has been shown to stabilize TAK1 and chaperones MAPK 4 and 7 [22]. Lastly, ERK5 signaling has been shown to be important in cardiac development and is slightly different from ERK1/2 [57,58]. Here, the cascade involves MEK5 which phosphorylates ERK5 which acts as a TF and activates other TFs including MEF2 which is implicated in cardiomyocyte hypertrophy [53]. ERK5 is found to be stabilized in the cytosol by Hsp90 [59]. Overall, Hsp90 is greatly involved in MAPK signaling and should be investigated further to modulate these pathways.

#### 2.1.3. PI3K/AKT(PKB)/mTOR Signaling

PI3K signaling is typically initiated by RTK or cytokine receptor activation [60]. Upon receptor activation, PI3K (p85 & p110) binds to the receptor via IRS and is phosphorylated. This complex phosphorylates phosphatidylinositol 4,5-bisphosphate (PIP2) which then phosphorylates phosphatidylinositol (3,4,5)-trisphosphate (PIP3). PIP3 activates PDK proteins which go on to phosphorylate protein kinase B (PKB) activating it. PKB acts as a kinase for many different proteins which control autophagy (mTOR), glucose metabolism (mTOR), protein synthesis (mTOR), proliferation, and cell survival [61]. It is clear that a major part of PI3K signaling consists of mTOR and its downstream targets. The mTOR protein is found in a complex with many others which aid in its function including RAPTOR in mTORc1. This complex inhibits ULK1 via phosphorylation thereby inhibiting autophagy [62]. It activates protein synthesis via p70S6K activation which activates S6, a ribosomal protein. It also inhibits 4E-BP1 which allows the elongation factors to form around the 5’ cap of mRNA [61]. Of these proteins, p85, p110, PKB, mTOR, RAPTOR, S6K, and eIF4E (translation elongation factor) are all Hsp90 clients [63,64,65]. It is also seen that inhibiting Hsp90 severely downregulates PKB and mTOR signaling [63,66]. There is also evidence that higher expression levels of Hsp90 preserve mitochondrial function through phosphorylation of Bcl2 in cardiomyocytes exposed to heat shock conditions via PKB and PKM2 signaling [67].

In the heart, it has been controversial whether this pathway is cardioprotective or cardiotoxic. Inhibiting PKB in the heart did not protect against hypertrophy and overexpressing PKB caused cardiac hypertrophy [68]. Modulation of PKB-dependent and PKB-independent pathways may have promise in protecting against IR injury however [69]. PI3K/PKB signaling may also be involved in non-canonical TGF-β signaling, suggesting crosstalk between the two pathways via p38 may play a role in pathology [70]. Altering mTOR signaling has also shown some promise in suppressing the inflammatory response in cardiomyocytes [71]. Since many of the proteins involved in the PI3K/PKB/mTOR pathway depend on Hsp90, perhaps it could be used as a target for modulating the pathway.

#### 2.1.4. G*_s_*/PKA Signaling and Calcium (Ca^2+^) Regulation

G protein coupled receptor (GPCR) signaling, especially *β*-adrenergic signaling, is essential in the heart and plays a critical role in the development of cardiomyopathy [72]. The activation of β-adrenergic receptors leads to phosphorylation of the heterotrimeric G-protein G_*s*_ which dissociates into Gα monomer and Gβ−γ dimer. The Gβ−γ dimer can activate potassium channels which allow an influx of potassium into the cell [73]. The Gα subunit stimulates adenylyl cyclase and subsequently activates protein kinase A (PKA) which has a wide range of activities including phosphorylation of sarcomere proteins [74,75,76] affecting contractility, regulation of RyR2 receptor which releases Ca^2+^ from the SR lumen [77] and phospholamban (PLN)/SR calcium ATPase (SERCA) which pumps Ca^2+^ back to the SR [78].

Hsp90 has been shown to mediate interactions between PLN, SERCA, and HAX-1. By recruiting Hsp90 to the SR Ca2+ uptake complex, the function of IRE-1, another Hsp90 client protein, was impaired [79,80,81]. Furthermore, the function of PLN and ryanodine receptor can be regulated by Ca^2+^/calmodulin-dependent protein kinase II (CaMKII) phosphorylation [78,82]. This kinase is also stabilized by Hsp90 [22]. CaMKII is relevant in intra-nuclear phosphorylation of transcription factors including HSF-1, CREB, and SRF. It also may activate NF-kB signaling leading to inflammatory response [83]. Given the role of CaMKII and SR Ca^2+^ cycling in the development of heart diseases, it is intriguing to examine if Hsp90 can be targeted to correct these dysfunctions. When Ca^2+^ levels increase in the cytosol, two important Ca^2+^-dependent proteins can be activated, calcineurin and calmodulin. Upon Ca^2+^ binding, these enzymes dimerize to form a functional phosphatase [84]. Hsp90 is found to stabilize both calcineurin and calmodulin and inhibition of Hsp90 leads to decreased nuclear factor of activated T-cells (NFAT) signaling [85,86]. Once the calcineurin/calmodulin phosphatase (CaM) is active, it dephosphorylates NFAT which is translocated to the nucleus as a transcription factor. Here, NFAT can activate genes controlled by MEF2 and GATA which are implicated in cardiac hypertrophy [87]. NFAT has been shown to be relevant in pathological cardiac hypertrophy and may also cross-talk with MAPK to accentuate pathological effects [88,89]. In addition to NFAT, CaM can also dephosphorylate and activate CaMKII, serving as another mechanism to potentiate CaMKII activity.

Calcium-related responses are not the only area of interest regarding β-adrenergic signaling. Desensitization of β adrenergic receptors plays a causal role in heart disease [90]. The common mechanism for β-adrenergic internalization depends on G protein coupled receptor kinases (GRKs), β-arrestin, and various proteins involved in endocytosis. GRKs are responsible for phosphorylating the receptor, thereby recruiting β-arrestin which initiates the internalization process. This recycling mechanism can also become dysregulated and β-adrenergic receptors can become desensitized by phosphorylation. Surprisingly, Hsp90 binds and stabilizes g protein-coupled receptor kinases GRK3, GRK5, and GRK6 [91] all of which are expressed in the heart [92]. Hsp90 also chaperones GRK2, allowing it to localize to the mitochondria. This promotes pro-death signaling in mice modeling I/R injury and in myocytes in vitro [93]. It has been found that GRK expression levels can drop as much as 70% when Hsp90 is inhibited [94]. This effect may be useful in targeting Hsp90 to prevent excess phosphorylation and desensitization of the β-adrenergic receptor, preventing development of cardiomyopathy and heart failure.

#### 2.1.5. G_*q*_/PKC Signaling

A different GPCR pathway is activated via angiotensin and endothelin receptors. The heterotrimeric g protein associated with these receptors is G_*q*_. Once activated through phosphorylation of the receptor, the α subunit goes on to activate protein lipase c (PLC), which ultimately activates protein kinase c (PKC) [95]. PKC has four isoforms in humans (α, β, δ, and ϵ) with α being the most abundant in heart [90]. Each of these isoforms has been found to have slightly different activity, for simplicity, this review will refer to all of them as PKC [96]. PKC has a wide range of targets which it phosphorylates. Some of the targets that are phosphorylated are sarcomere proteins l which will alter the stiffness of the myocardium and can contribute to the onset of cardiomyopathy if dysregulated [96]. PKC affects phospholamban (PLN) indirectly through phosphorylation of I-1, this inhibits PP1 which directly regulates PLN [97]. This leads to a decrease in phosphorylation causing a decrease in Ca^2+^ uptake by SERCA2 and cardiac dysfunction [98]. There is also crosstalk between PKC and MAPK through ERK1/2 which implicates PKC in the expression of hypertrophy-related gene expression [99]. There is also evidence of PKC activating NF-kB in cardiomyocytes, causing expression of pro-inflammatory proteins implicated in fibrosis [100]. Hsp90 is known to regulate NF-kB through stabilization of IkB kinase [101]. Lastly, PKC can also be cleaved by calpain (a Ca^2+^ dependent protease) which is stabilized by Hsp90 in the cytosol. This cleavage makes a fragment called PKMα which is implicated in dilated cardiomyopathy [102].

#### 2.1.6. TNFα Signaling

Tumor necrosis factor α (TNF-α) signaling is known to activate apoptosis, necrosis, proliferation, and inflammation responses in cells. In the heart, this type of signaling is relevant in myocardial remodeling and is typically induced in myocytes by IR injury and HF [103]. Initially, the cytokine TNFα binds to its receptors, TNFR1 or TNFR2. Both receptors are expressed in the heart and are upregulated following IR injury [104]. TNFR1 signaling is more associated with apoptosis and necrosis response, while TNFR2 response results in proliferative and inflammatory genes, suggesting TNFR1 is cardiotoxic and TNFR2 is cardioprotective in response to injury [105]. In both pathways, many of the signaling proteins are stabilized by Hsp90.

TNFR1 signaling can be apoptotic, necrotic, and inflammatory. This type of response by TNFR1 is mediated by TRADD and receptor-interacting serine/threonine-protein kinase 1 (RIPK1). TRADD binds the receptor and recruits RIPK1. RIPK1 is phosphorylated then dimerizes. This active form of RIPK1 can then form two different complexes, one being the necrosome, the other being apoptotic. The necrosome consists of RIPK1, RIPK3, FADD, MLKL, Casp8, and ubiquitin. All of these proteins besides FADD are known Hsp90 clients [106,107,108]. In fact, the formation of the necrosome has been found to be dependent on Hsp90, showing that Hsp90 plays a regulatory role in TNFα-induced necrosis [108]. Following the activation of MLKL, it polymerizes and forms a cation pore in the membrane [109]. This influx of cations results in ER stress causing a spike in reactive oxygen species, leading to the opening of the mitochondrial permeability transition pore (mPTP) in cardiomyocytes [110]. The mPTP is regulated by cyclophilin D, which is also stabilized by Hsp90 [80], showing another level of necrosis regulation by Hsp90. In the apoptotic response, RIPK1, FADD, TRADD, and procaspase 8 form a complex where procaspase 8 becomes activated and triggers apoptosis. Again, Hsp90 is found to bind all of these proteins except FADD [22] and can play a role in regulation [111].

TNFR2 signaling is most commonly associated with inflammation. TNFα binds TNFR2 and a complex forms around the intracellular portion of the receptor, similar to TNFR1. However, there are differences in the proteins recruited [112]. The TNFR2 forms a complex with multiple proteins, including TAK1, IKK2 (IKKa/IKKb), and NEMO, which are crucial in the initiation of NF-kB. Hsp90 is required for the recruitment of IKK2 to the receptor [113] and is also important in IKKa/IKKb stabilization in cardiomyocytes [101]. Studies have shown that treatment with geldanamycin (Hsp90 inhibitor) disrupts TNFα induced NF-kB signaling [106,114]. NF-kB is a prevalent pro-inflammatory pathway and the activation of this transcription factor leads to expression of pro-inflammatory cytokines which play roles in cardiac hypertrophy, fibrosis, and repair following IR injury.

### 2.2. Pathophysiological Significance in Cardiomyopathy

Hsp90 plays a role in all of the aforementioned pathways by stabilizing or folding various proteins in each cascade. These pathways have all been implicated in pathological fibrosis, hypertrophy, cell death responses in the heart and the effects of inhibiting Hsp90 have been studied in each as well (Table 1). Hsp90 in the context of the TGF-β pathway has been studied due to the pro-fibrotic gene expression it causes in the heart. It has been found that inhibiting Hsp90 using either geldanamycin or an inhibitory peptide prevents pro-fibrotic TGF-β signaling in cardiomyocytes and cardiac fibroblasts [45,115].

The effects of Hsp90 inhibition on the MAPK pathway has also been studied recently. Rats treated with 17-AAG ((17-(allylamino)-17-dimethoxy-geldanamycin) two weeks after undergoing coronary artery ligation survived better and maintained better cardiac function compared to rats not receiving 17-AAG treatments [117]. This shows a direct link between Hsp90 function, MAPK signaling, and development of cardiomyopathy following I/R injury. The PI3K/AKT/mTOR pathway and Hsp90 were studied following heat shock damage in vitro. Here, it was seen that Hsp90 inhibition in mice using geldanamycin prevented the cardioprotective action of AKT under heat shock conditions leading to more apoptosis [67]. While this may not be a direct link to cardiomyopathy, it shows that Hsp90 and AKT may protect against apoptosis in the heart.

In the G_*s*_/PKA pathway, Hsp90 is found to be involved in the regulation of SERCA2 via HAX-1, which has implications in arrhythmia in cardiomyocytes [80,81]. In the G_*q*_/PKC pathway, there is one study that shows hypertrophic angiotensin II was prevented with administration of geldanamycin. This occurs from Hsp90’s role in stabilizing the IKK complex which is required for NF-kB. Inhibiting this signaling pathway using geldanamycin prevents hypertrophic signaling in cardiomyocytes [101].

Lastly, the TNF-α pathway was demonstrated to be affected by geldanamycin treatment in ischemic postconditioning. In rats, the postconditioning treatment was able to reduce infarct size partially through reducing TNF-α signaling. Rats that underwent postconditioning and given geldanamycin saw the infarct size return to the same levels as rats that received no postconditioning treatment showing that the inhibition of Hsp90 increases TNFα via JNK signaling [122]. Necrosis in mouse heart has also been shown to be regulated by Hsp90 via HAX-1, cyclophilin D, and mPTP [121]. Hsp90 also plays a cardioprotective regulatory role in apoptosis through HAX-1 and IRE-1 in mouse heart. Here, inhibition of Hsp90 prevented additional cardioprotective effects of HAX-1 [79]. These studies demonstrate that the modulation of Hsp90 can affect any one of these pathways which may be able to modulate physiological characteristics seen in diseased states (Figure 2). More importantly, these studies also exemplify the challenge in developing Hsp90 therapy, as it is not completely clear if inhibiting Hsp90 can benefit the heart under each of the pathological conditions. Indeed, systemic inhibition of Hsp90 is associated with development of cardiomyopathy [123,124,125,126]. Thus, instead of affecting every Hsp90 complex in the cardiac cells, developing a strategy to target a subset of Hsp90 client protein may be a better approach.

## 3. Hsp90 Modulation

### 3.1. Inhibitors and Agonists

For decades, Hsp90 inhibitors have been studied for their potential in treating cancers, leading to the discovery and development of nearly 90 inhibitors with several entering clinical trials [127,128]. The cumulative knowledge from studying Hsp90 inhibition may allow Hsp90 modulation for treating cardiac disease. Conventional inhibitors halt Hsp90 ATPase function by binding to the NTD. This prevents the main chaperone function and potentially leads to the degradation of client proteins. Failed protein folding can lead to Hsp90 presenting clients to the appropriate ubiquitin ligase, and eventually deconstructed by the ubiquitin-proteasome system [129]. NTD inhibitors can have a considerable effect on the total native structure of client proteins. Categories of these include aryltriazine, amide, resorcinol, and primidine-diamine derivatives. Some cause pan-inhibition of Hsp90 while others only inhibit specific isoforms [130].

More recent is the continued study of CTD inhibitors, which typically exhibit allosteric control of the N-terminal binding site. The C-terminal binding interferes with client protein folding and co-chaperone activity, leading to similar downstream effects [131]. Various triazole modified coumarin analogs were synthesized from the inspiration of NTD inhibitor novobiocin. Further modifications include replacing the coumarin skeleton with a biphenyl amide backbone, and introducing aromatic rings.

Other small molecules can also modulate Hsp90. From synthetic peptides to natural products these molecules can bind to different regions of Hsp90 for inhibition or agonization. There is also an effort to design peptides that increase the specificity of client proteins, avoiding some of the adverse effects of chemical inhibition [130]. For example, a peptide inhibitor to target the N-terminal helix-loop-helix domain of Grp94 was designed to suppress specifically LPS responses [132]. The drug KU-32 (a novobiocin derivative) increased chaperone activity by binding to the CTD and forcing the NTD into a partially closed intermediate phase that increases ATPase activity [133]. Drug KU-596 also increases HSP levels by binding to the Hsp90 CTD [134].

Another way to modulate Hsp90 activity is with molecules that increase its transcription. As discussed earlier, the transcription of Hsp90α is mediated by heat shock factors. Of the six HSFs encoded in the human genome (HSF1, HSF2, HSF4, HSF5, HSFX, HSFY), HSF1 plays the greatest role in Hsp90 transcription [131]. During normal cellular conditions, HSF-1 monomers reside compelxed with Hsp40, Hsp70, Hsp90, and chaperonin TCP1 ring complex. During heat shock, or other stressors, HSF1 dissociates, trimerizes, and undergoes transport to the nucleus. During its active cycle, HSF1 can be heavily influenced by posttranslational modifications (PTMs). In general, acetylation promotes HSF1 stability and phosphorylation causes degradation [131]. As a part of negative feedback, Hsp90 can bind to HSF1 to repress its activity and lower Hsp90 transcription. Because of this, low levels of certain Hsp90 inhibitors can actually induce general heat shock response by locally elevating HSF1 levels. For instance, geldanamycin, radicicol, and celastrol [135]. In some instances, this is done at significantly lower concentrations than if the drug’s purpose was to degrade client proteins [136]. Various other compounds can also activate HSF1 and lead to Hsp90 transcription. Protein translation inhibitors (such as puromycin), amino acid analogs (such as azetidine 2-carboxlate and canavaine), thiol-reactive molecules, and some organic electrophiles can all activate the heat shock response [135]. This combination of Hsp90 inhibitors and heat shock response activators creates a considerable ensemble for modulating Hsp90 activity in the cell.

### 3.2. Post-Translational Modifications

While inhibition of Hsp90 is a proven method to treat cancer, systemic inhibition of Hsp90 comes with side effects including development of cardiomyopathy [123,124,125,126]. This brings a basic research question: Could a more specific targeting of Hsp90 via modulation of PTMs or PTM-causing enzymes be a revolutionary therapeutic method for a variety of diseases where Hsp90 plays a role including cardiomyopathy? It is reported that there are more than 400 different types of PTMs found in eukaryotic cells [137]. Hsp90 is subject to many posttranslational modifications and these PTMs have been extensively reviewed in [138,139]. This section aims to discuss cardiac-related PTMs of Hsp90 and how a few have been shown to play a role in diseases (Table 2).

PTMs can change activities of Hsp90 through covalent modification and addition of functional groups or proteins. The addition of these groups can cause conformational change and alter protein–protein interactions. The main PTMs seen in Hsp90 are phosphorylation, acetylation, methylation, nitrosylation, SUMOylation, and ubiquitination [138]. Phosphorylation is thought to regulate chaperone activity of Hsp90 since hyperphosphorylation is associated with decreased ATPase activity [139]. There are multiple phosphorylation sites on Hsp90 many of which are seen to downregulate ATPase activity. Acetylation is also prevalent in Hsp90 but occurs in a smaller amount than phosphorylation. These acetylations are shown to decrease Hsp90 ATPase activity much the same as phosphorylation does [138]. The other PTM types mentioned before have much less data available due to lack of experiments done on their effects. They have mainly been found through high-throughput proteomic-based analyses. A few groups have been able to identify specific PTMs and their potential roles in cardiomyopathy [138,139].

There have been a few studies supporting the hypothesis of Hsp90 PTMs being used as a therapeutic approach. Methylation of Hsp90 at lysine 616 (K616) by Smyd2 directly affects Hsp90–sarcomere interactions. Without methylation at K616, sarcomere proteins are less stable and are degraded more easily compared to Hsp90 without that methylation [140]. Another study found nitrosylation of Hsp90 at cystine 589 (C589) by nitric oxide (NO) results in further stabilization of the TGF-β receptor in cardiomyocytes. When this site was mutated to prevent nitrosylation, TGF-β receptor was degraded more readily and profibrotic signaling was downregulated [141]. Outside of these two studies, cell types other than cardiomyocytes were used to show effects of Hsp90. They have shown that phosphorylation at T89 and T36 can prevent androgen receptor client interaction (T89) and CDC37-Hsp90 complex formation (T36) [142,143]. Without interaction with CDC37, Hsp90 would not be able to serve as a chaperone for kinase clients and would greatly affect pathways like MAPK, CaM, and CaMKII. Another study found SUMOylation at lysine 191 (K191) is responsible for recruiting Aha1 [139]. This is vital because Aha1 drives chaperone activity of Hsp90 by upregulating ATPase activity. Inhibition of this PTM would likely inhibit Hsp90 activity. Lastly, acetylation of Hsp90 at lysine 294 (K294) shows reduced client and co-chaperone interactions [144]. Without consistent co-chaperone binding and client recruitment, Hsp90 activity would be downregulated, having potential effects on all pathways described prior. The ideal way forward is to design experiments to test the effects of Hsp90 PTMs in cardiomyocytes to determine new therapeutics that modulate Hsp90 activity that extends to pathways described above.

## 4. Summary

Understanding the Hsp90 interactome can help uncover the role it may play in these pathways, and how those roles could be changed to improve patient outcome in cardiomyopathy. Although it is without a doubt that Hsp90 is highly involved in these pathways through chaperone activity and stabilization that prevents degradation of signaling proteins, its numerous substrates pose a difficult task to fully understand the role of Hsp90 in each pathologic condition. While the effects of inhibiting Hsp90 have not been investigated in every context, many studies have investigated them, and can demonstrate the effect of inhibition on certain pathways. Perhaps combining our understanding of the effects of Hsp90 inhibitors with the growing knowledge of Hsp90 PTMs and their effects could provide a new type of therapeutic course through modulation of Hsp90 activity rather than systemic inhibition. In this regard, the tremendous advancement in omic analysis and model prediction should bring us closer to fully realize the potential of targeting Hsp90 in cardiac diseases. We anticipate that more efforts will be placed to dissect the alteration on Hsp90-related pathways, which will provide us with a better picture of cellular dysfunctions that can be targeted by Hsp90 treatments.

## Figures and Tables

**Figure 1 jpm-11-01373-f001:**
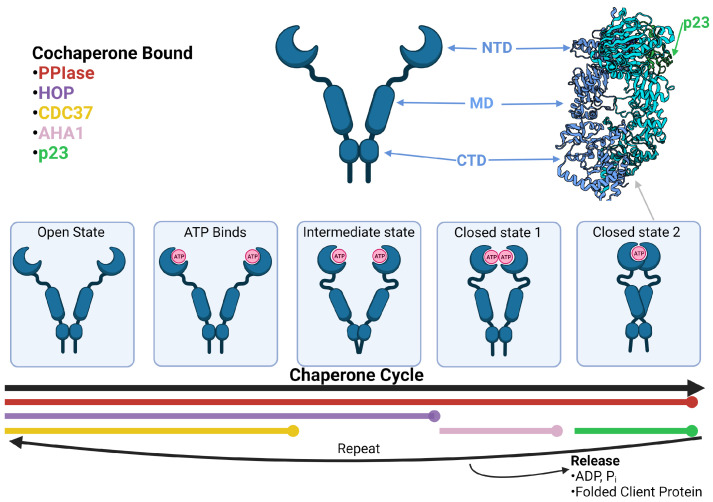
Hsp90 structure and chaperone cycle. The Hsp90 homodimer goes through multiple conformational changes while folding client proteins. The phases when certain cochaperones, PPIase, HOP, CDC47, AHA1, and p23, bind are labeled by color. In the top right is the Cryo-EM structure of Hsp90 complexed with p23 in closed state 2 (PDB ID: 7L7J from [32]). The domains of Hsp90 are labeled NTD, MD, and CTD.

**Figure 2 jpm-11-01373-f002:**
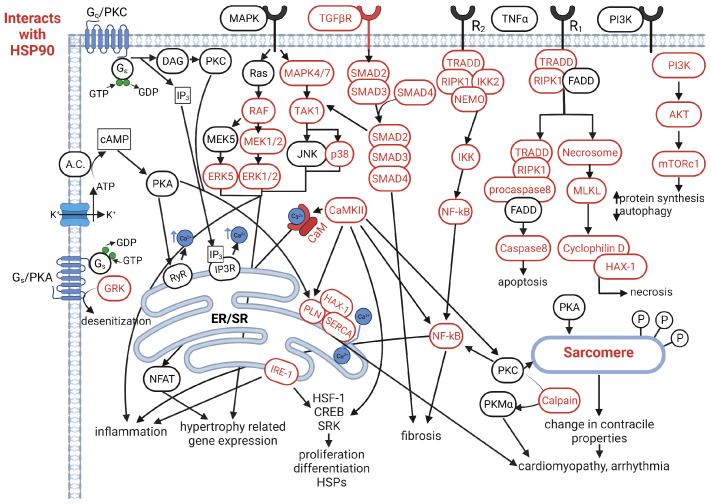
Signaling pathways related to cardiomyopathy. Proteins highlighted in red interact with Hsp90.

**Table 1 jpm-11-01373-t001:** Summary of Hsp90 involvement and effects of inhibition of Hsp90 on the aforementioned pathways as well as the phenotypic result of modulating Hsp90 in the heart. Evidence for Hsp90 involvement in pathophysiology can be found, however, it is a relatively new approach to investigating cardiomyopathy so studies are limited.

Pathway	Hsp90 Effect on Pathway	Cardiac Phenotype	Reference
**TGF-β**			
	Protect TGF-β receptor from degradation allowing fibrosis signaling.	myocardial fibrosis	[44,45]
	Inhibition prevents TGF-β fibrotic signaling.	myocardial fibrosis	[115,116]
**MAPK**			
	Stablize TAK1 and p38 to allow hypertrophic signaling.	hypertrophic responses and apoptosis	[46,47]
	Inhibition protects rats from cardiac hypertrophy and failure by inhibiting MAPK signaling.	hypertrophic response	[117]
	Inhibition prevents c-Raf-Erk induced fibrosis.	hypertrophic response and myocardial fibrosis	[118]
**PI3K/AKT/mTOR**			
	Hsp90 overexpression in cardiomyocytes preserves mitochondrial function via AKT and PKM2 signaling in vitro.	apoptosis	[67]
	Inhibition increases apoptosis in cardiomyocytes under hypoxia conditions.	apoptosis	[119]
**G_*s*_/PKA**			
	Mediates PLN, SERCA2, and HAX1 interactions to affect Ca^2+^ signaling and contractility.	cardiac contractility	[80]
	Inhibition prevents calcineurin-NFAT-induced fibrosis.	hypertrophic response and myocardial fibrosis	[119]
**G_*q*_/PKC**			
	Inhibition causes reduced angiotensin II-induced hypertrophy and NF-kB signaling.	hypertrophic response	[101]
	Stabilizes caplain which can cleave PKC to PKMα. PKMα can cause dilated cardiomyopathy.	dilated cardiomyopathy	[102,120]
**TNFα**			
	Stabilizes cyclophilin D to regulate cell death via mPTP.	necrosis	[80,121]
	Required for TNFα induced NF-kB signaling.	hypertrophic response	[101,113]
	HAX-1 binds to Hsp90 to mediate cardioprotection.	necrosis and apoptosis	[79,121]

**Table 2 jpm-11-01373-t002:** This table summarizes Hsp90 PTMs. The headers of the table are as follows: “Enzyme” is the enzyme that does the PTM. “PTM” is the type of modification. “Hsp90 Site” is this amino acid location of the modification. “Effect” describes experimental findings of the PTM.

Source	Enzyme	PTM	Hsp90 site	Effects	Pathology
[140]	Smyd2	Methylation	K616	methyl-hsp90 needed in proper sarcomere function	arrhythmia and myocardial stiffening
[141]		Nitrosylation	C589	sno-hsp90 stablizes TGF-β receptor and promotes fibrosis	DCM, HCM
[142]	PKA	Phosphorylation	T89	reduces Hsp90-androgen receptor interaction	N/A
[143]	CK2	Phosphorylation	T36	inhibition of cdc37-hsp90 complex	N/A
[139]		SUMOylation	K191	recruits Aha1	N/A
[144]		Acetylation	K294	reduced client and cochaperone interactions	N/A

## Data Availability

Not applicable.

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
