# Peer review of "The Potential of Hsp90 in Targeting Pathological Pathways in Cardiac Diseases"

_jpm, 2021, doi:10.3390/jpm11121373_

Round 1
Reviewer 1 Report
Roberts, Hallee and Lam are aiming to give a interesting perspective on the topic of chapperones in cardiomyopathy (development) for therapeutic targets. The authors give a description of Hsp90 and its global function, followed by a thin explanation of cardiomyopathies that is merely insufficient and partly incorrect. The main body of this literature review then continues with summarizing the interaction between Hsp90 and different molecular pathways. Very little connection to heart disease is made hereafter. In conclusion, the content does not reflect the title of the manuscript. Below I have listed my concerns regarding the quality and readability of the manuscript:
Major:
Line 36 and 37: “Hsp90, Hsp90” >> should be alpha and beta. The authors use Hsp90, Hsp90 as two different isoforms in line 36 and 37, which is confusing. The article in 2019 herafter is using a more clear strategy: the four Hsp90 isoforms (Hsp90α/β, Grp94, TRAP1) >> Please add Ref: https://www.frontiersin.org/articles/10.3389/fnmol.2019.00294/full.
Line 39, 40 and 42: same issue, I’m afraid the alpha and beta symbols went missing.
Line 44-45: “In this review, we will focus on the cytosolic Hsp90 and the name Hsp90 will refer to both Hsp90 and Hsp90 unless specified otherwise.” >> I now cannot follow this anymore. I think the Hsp90 isoforms should be specified to avoid that readers which jump in to certain areas (directed by search-engines such as Google are drawing the wrong conclusions).
Line 66-97: This part is very important for the understanding of the function of a chaperone. The authors, however, use may difficult terms and also throw in extra information, such as affinity for steroids, or clients. Also, to me it is not clear if with cell division and maturation, the authors refer to these actual biological cellular processes. I think for mostly uninformed readers that are targeted by this review of Hsp90 in CMP, I advise to simplify this part or use extra figure annotations to help the reader understand this part more.
Line 99: “Cardiomyopathy (CM)” >> the common abbreviation for cardiomyopathy is CMP
Line 100: the most common form of CMP is missing, namely ischemic CMP.
Line 101: if the authors decide to come up with a 4-way split of genetic/inherited CMPs, then arrhythmogenic (ARVC) should be changed to ACM. ARVC refers to a specific form of arrhythmogenic right ventricular cardiomyopathy (mostly seen in young male adolescents/adults). The group of channelopathy suchs as Brugada (SCN5A) are forgotten this way. I think the authors should read in more, or make it simpler by saying inherited cardiomyopathies such as DCM, HCM, RCM and ACM (not mentioning that there are only four).
Line 104: The end result of CM in many cases is heart failure (HF) >> incorrect. The result is heart failure (any decrease in left ventricular function is called heart failure). The end-stage is called congestive heart failure.
Line 104: HF places >> The clinical syndrome of HF places
Line 106-108: “Interestingly, Hsp90 plays an important role in many of these disease state pathways related to CM, those primarily being hypertrophy, fibrosis, necrosis, and apoptosis responses.” >> This is a bold statement and the authors cannot state this in the current manner. It should be either weakened or directly supported with evidence (at least 5 references to scientific papers that support this message).
Line 406-407: “We anticipate that more efforts will be placed to dissect the alteration on Hsp90-related pathways, which will give us a better picture of cellular dysfunctions that can be targeted by Hsp90 treatments.” >> Personally, I’m missing the efforts of the authors to connect the role/function of HSP90 to the heart. The build-up of the rest of the manuscript should be changed to the different and/or overlapping roles of HSP90 which have been found/reported in each form of CMP: DCM, HCM, RCM, ACM and Ischemic. This set up would suit the title of the manuscript in a much better way.
Minor:
Line20: 20-3% >> 20-30%?
Line 31: Here, the focus of this review will be on Hsp90 >> Here we focus on Hsp90 or This review is focused on Hsp90.

Author Response
Roberts, Hallee and Lam are aiming to give a interesting perspective on the topic of chapperones in cardiomyopathy (development) for therapeutic targets. The authors give a description of Hsp90 and its global function, followed by a thin explanation of cardiomyopathies that is merely insufficient and partly incorrect. The main body of this literature review then continues with summarizing the interaction between Hsp90 and different molecular pathways. Very little connection to heart disease is made hereafter. In conclusion, the content does not reflect the title of the manuscript. Below I have listed my concerns regarding the quality and readability of the manuscript:
Response:
A huge thanks to the reviewer for their thoughtful insight. We have addressed the major and minor comments below. Given the confusion on the use of cardiomyopathy in the previous manuscript, we changed the title to “The Potential of Hsp90 in Targeting Pathological Pathways in Cardiac Diseases”, which fits more to our intention to describe the effect of Hsp90 in various subcellular pathological processes. Moreover, we included new sections and a new table to address the connection between Hsp90 and cardiac diseases. Overall, thanks to the suggestion of this reviewer, the quality of this review article is drastically improved. Specific response to each concern is provided below.
Major:
Line 36 and 37: “Hsp90, Hsp90” >> should be alpha and beta. The authors use Hsp90, Hsp90 as two different isoforms in line 36 and 37, which is confusing. The article in 2019 herafter is using a more clear strategy: the four Hsp90 isoforms (Hsp90α/β, Grp94, TRAP1) >> Please add Ref: https://www.frontiersin.org/articles/10.3389/fnmol.2019.00294/full.
Line 39, 40 and 42: same issue, I’m afraid the alpha and beta symbols went missing.
Line 44-45: “In this review, we will focus on the cytosolic Hsp90 and the name Hsp90 will refer to both Hsp90 and Hsp90 unless specified otherwise.” >> I now cannot follow this anymore. I think the Hsp90 isoforms should be specified to avoid that readers which jump in to certain areas (directed by search-engines such as Google are drawing the wrong conclusions).
Response:
Thanks to the reviewer for pointing out this formatting issue in our manuscript. The alphas and betas were not formatted correctly, as it was our first time to use the LaTeX format for submission. They have now been added to our manuscript (lines 29-49). We also added the requested reference to line 37 (reference #6) and we have modified the text to follow this labeling strategy. Furthermore, we removed the problematic sentence on the original lines 44-45 (indicated on lines 48-49 in the manuscript with changes tracked). We hope that these changes ensure the isoforms are strictly distinguished.
Line 66-97: This part is very important for the understanding of the function of a chaperone. The authors, however, use may difficult terms and also throw in extra information, such as affinity for steroids, or clients. Also, to me it is not clear if with cell division and maturation, the authors refer to these actual biological cellular processes. I think for mostly uninformed readers that are targeted by this review of Hsp90 in CMP, I advise to simplify this part or use extra figure annotations to help the reader understand this part more.
Response:
We thank the reviewer for raising the concern. To reduce the confusion, we removed the description of steroid hormone receptors. In the field of Hsp90, client, or client protein, is the universal term to describe the substrate that will be folded with Hsp90 assistance. We particularly added a small description to address this on line 58 in the section of 1.2 Hsp90 Structure. We used cell division, or other cellular process-related terms, because they are part of the full name of the protein, such as CDC37 (cell division cycle 37). In order to clarify it, the abbreviation was used first, with the full name in parenthesis (lines 74-77, 82-86, 90-91, 94-97 in the manuscript with changes tracked). We hope that these changes will help the reviewer and readers to understand this part easier.
Line 99: “Cardiomyopathy (CM)” >> the common abbreviation for cardiomyopathy is CMP
Response:
We thank the reviewer for the suggestion. To avoid confusion, we decided not to use abbreviation when we used the word “cardiomyopathy” alone. When it is coupled with dilated, hypertrophic, arrhythmogenic and restrictive, we will instead use DCM, HCM, ACM and RCM, which are commonly accepted abbreviations.
Line 100: the most common form of CMP is missing, namely ischemic CMP.
Response:
Thank the reviewer for pointing it out. Description of ischemic cardiomyopathy has been included on lines 112-113.
Line 101: if the authors decide to come up with a 4-way split of genetic/inherited CMPs, then arrhythmogenic (ARVC) should be changed to ACM. ARVC refers to a specific form of arrhythmogenic right ventricular cardiomyopathy (mostly seen in young male adolescents/adults). The group of channelopathy suchs as Brugada (SCN5A) are forgotten this way. I think the authors should read in more, or make it simpler by saying inherited cardiomyopathies such as DCM, HCM, RCM and ACM (not mentioning that there are only four).
Response:
We thank the reviewer for the suggestion. ARVC was changed to ACM, since we unintentionally narrowed our scope by discussing ARVC. More reading was done to further our understanding. Each cardiomyopathy has been left with minimal description since the paper does not focus on the different types of cardiomyopathy. Instead, we opted to mention their existence and typical causes, alluding to the fact that the pathways being discussed afterwards play roles in the development of cardiomyopathies. We have also distinguished DCM, HCM, RCM, and ACM as inherited while ischemic cardiomyopathy occurs mainly through lack of oxygen to the myocardium via coronary artery disease and atherosclerosis. (please refer to line 112-118)
Line 104: The end result of CM in many cases is heart failure (HF) >> incorrect. The result is heart failure (any decrease in left ventricular function is called heart failure). The end-stage is called congestive heart failure.
Line 104: HF places >> The clinical syndrome of HF places
Response:
We thank the reviewer for pointing out the mistake. Original line 104 has been reworded to account for this distinction. The requested changes have been made on lines 118-119.
Line 106-108: “Interestingly, Hsp90 plays an important role in many of these disease state pathways related to CM, those primarily being hypertrophy, fibrosis, necrosis, and apoptosis responses.” >> This is a bold statement and the authors cannot state this in the current manner. It should be either weakened or directly supported with evidence (at least 5 references to scientific papers that support this message).
Response:
We sincerely thank the reviewer for the constructive feedback. To help support the claim from original lines 106-108, a new section and new table were added to this paper. The new section provides evidence for Hsp90 inhibition affecting the pathways mentioned in section 2.2 “Pathophysiological Significance in Cardiac Diseases”. These Hsp90 inhibition studies were performed in vivo (mice or rats) and in vitro (cardiomyocytes or cardiac fibroblasts). This new section aims to solidify the claim from lines 121-127 from the first version of the manuscript. All of these findings are also summarized in the new Table 1. The table has the headers: Pathway, Hsp90 Effect on Pathway, Cardiac Phenotype, and References. The “Pathway” column is the name of the pathway described in section 2. The “Hsp90 Effect on Pathway” section gives a 1-2 sentence description of the Hsp90-related findings of the paper. The “Cardiac Phenotype” section provides the pathological response associated with the findings from the paper. The “References” column provides the citation number so readers can easily find the information if they would like to.
Moreover, this paragraph serves to introduce the pathological pathways regulated by Hsp90 that are implicated in cardiac diseases development. Therefore, all supporting literature is described in the following subsections, such as TGFbeta-signaling, MAPK signaling.. etc. Thus, we hope that the pre-existing section on each Hsp90 regulated pathway, a new section on the connection between Hsp90 and cardiac diseases, and a new table to summarize all related literature can alleviate reviewer’s concern on the claim.
Line 406-407: “We anticipate that more efforts will be placed to dissect the alteration on Hsp90-related pathways, which will give us a better picture of cellular dysfunctions that can be targeted by Hsp90 treatments.” >> Personally, I’m missing the efforts of the authors to connect the role/function of HSP90 to the heart. The build-up of the rest of the manuscript should be changed to the different and/or overlapping roles of HSP90 which have been found/reported in each form of CMP: DCM, HCM, RCM, ACM and Ischemic. This set up would suit the title of the manuscript in a much better way.
Response:
We thank the reviewer for raising the concern. Since current findings are insufficient to accommodate the suggestion from the reviewer to discuss the role of Hsp90 in each form of cardiomyopathy, we decided to change the title to “The Potential of Hsp90 in Targeting Pathological Pathways in Cardiac Diseases”. We believe the title suits better with the current manuscript setup to target audience in basic cardiac research. Moreover, we also believe that this change can avoid confusion over the intention of the current manuscript to discuss how Hsp90 regulates pathological pathways in cardiac diseases.
Furthermore, to provide a better picture of how Hsp90 can affect cardiomyopathy development, we generated a new section 2.2 named “Pathophysiological Significance in Cardiac Diseases” (lines 315-355) to summarize the physiological aspects of Hsp90 inhibition and its effects on the aforementioned pathways from the second section. Table 1 has also been added which highlights literature showing Hsp90 involvement and effects on the pathway. By adding a new section and a new table, we hope that readers will see the connection between Hsp90 and cardiac diseases more easily. Furthermore, we believe the table will also serve as an important tool for readers to look for Hsp90 related literature in the cardiac field.
Minor:
Line20: 20-3% >> 20-30%?
Line 31: Here, the focus of this review will be on Hsp90 >> Here we focus on Hsp90 or This review is focused on Hsp90.
Response:
Thanks for pointing them out. The requested changes were made on lines 20 and 31 respectively.
Reviewer 2 Report
Overall, this is a good review. However, there are some mistaken sentences that need to be corrected in this paper, otherwise it will mislead the readers, such as “Hsp90 expression is inducible and regulated by heat shock factor 1 (HSF1), while Hsp90 is constitutively expressed [7]. Together, these two isoforms make up 1-2% of cytosolic proteins in normal homeostasis, and up to 4-6% when a cell is stressed [8]. It has been suggested that Hsp90 and Hsp90 together potentially interact with 10% of all cytosolic proteins [9] which demonstrates how important it is to understand the roles they play. In this review, we will focus on the cytosolic Hsp90 and the name Hsp90 will refer to both Hsp90 and Hsp90 unless specified otherwise”. Here, Hsp90alpha and Hsp90beta should be strictly distinguished.

Author Response
Overall, this is a good review. However, there are some mistaken sentences that need to be corrected in this paper, otherwise it will mislead the readers, such as “Hsp90 expression is inducible and regulated by heat shock factor 1 (HSF1), while Hsp90 is constitutively expressed [7]. Together, these two isoforms make up 1-2% of cytosolic proteins in normal homeostasis, and up to 4-6% when a cell is stressed [8]. It has been suggested that Hsp90 and Hsp90 together potentially interact with 10% of all cytosolic proteins [9] which demonstrates how important it is to understand the roles they play. In this review, we will focus on the cytosolic Hsp90 and the name Hsp90 will refer to both Hsp90 and Hsp90 unless specified otherwise”. Here, Hsp90alpha and Hsp90beta should be strictly distinguished.
Response:
Thanks to the reviewer for pointing out this formatting issue in our manuscript, we greatly appreciate it. The alphas and betas were not formatted correctly as it was our first time to use the LaTeX format for submission. They have now been added to our manuscript (lines 29-49). We hope that this change ensures the isoforms are strictly distinguished.
Reviewer 3 Report
This was a well written and informative review that brings attention to a heavily researched protein regarding potential opportunities and strategies for its therapeutic targeting the management of heart disease. The authors go into great detail regarding the mechanisms by which Hsp90 interacts with a diverse array of signaling pathways and the impact those signaling pathways have on cardiac disease remodeling. However, before publication, it would be helpful to see a bit more discussion on the direct impacts of Hsp90 on cardiac remodeling, should this data be available, and provide conceptual understanding that targeting Hsp90 can alter the course of cardiac remodeling (some references listed in this regard).
Comments:
- When discussing hsp90 isoforms hsp90 is used twice when referring to for different isoforms (line 36 and 37). This is consistently done throughout the manuscript. It is not clear if the author meant to say Hsp90α and Hsp90β when naming these multiple isoforms early on.
- Of the current drugs available to treat hsp90 in cancer, is there data available to suggest associations in consequences on cardiotoxicity
- In lines 368-369 PTMs are identified with a brief discussion on impact to Hsp90 activity. Are there any known associations of Hsp90 PTMs that associate with different states of cardiac disease that can be described or added to table 1.
- A bit more discussion of known modulation of hsp90 and cardiac outcomes would be beneficial (ex PMIDs: 30582930 and 29203715). Also, a discussion of any known mutations for gain- or loss-of-function (if there are any) in hsp90 and the impact on the heart.
Author Response
This was a well written and informative review that brings attention to a heavily researched protein regarding potential opportunities and strategies for its therapeutic targeting the management of heart disease. The authors go into great detail regarding the mechanisms by which Hsp90 interacts with a diverse array of signaling pathways and the impact those signaling pathways have on cardiac disease remodeling. However, before publication, it would be helpful to see a bit more discussion on the direct impacts of Hsp90 on cardiac remodeling, should this data be available, and provide conceptual understanding that targeting Hsp90 can alter the course of cardiac remodeling (some references listed in this regard).
Response:
We sincerely thank the reviewer for the encouragement and have made changes to the manuscript according to your comments, detailed below.
When discussing hsp90 isoforms hsp90 is used twice when referring to for different isoforms (line 36 and 37). This is consistently done throughout the manuscript. It is not clear if the author meant to say Hsp90α and Hsp90β when naming these multiple isoforms early on.
Response:
Thanks to the reviewer for pointing out this formatting issue in our manuscript. The alphas and betas were not formatted correctly, as it was our first time to use the LaTeX format for submission. They have now been added to our manuscript (lines 29-49). We hope that this change ensures the isoforms are strictly distinguished.
Of the current drugs available to treat hsp90 in cancer, is there data available to suggest associations in consequences on cardiotoxicity.
Response:
We thank the reviewer for raising this question. There are indeed literatures reporting cardiotoxicity associated with the use of Hsp90 inhibitors. Three studies have been added to the citation for the claim “systemic inhibition of Hsp90 comes with side effects including development of cardiomyopathy.” Two of these are clinical trials of Hsp90 inhibitors which had shown cardiotoxicity side effects (doi: 10.1038/leu.2009.292, doi: 10.3109/10428194.2012.760733). The third is a study that found hERG channels in the heart are chaperoned by Hsp90 which raises concerns about inhibition causing side effects in cardiac function (doi: 10.1161/01.RES.0000079028.31393.15). Corresponding description has been added to section 2.2 “Pathophysiological Significance in Cardiac Diseases” at lines 352-355 and 3.2 “Post-Translational Modifications” at lines 405-407.
In lines 368-369 PTMs are identified with a brief discussion on impact to Hsp90 activity. Are there any known associations of Hsp90 PTMs that associate with different states of cardiac disease that can be described or added to table 1.
Response:
We thank the reviewer for the suggestion. A column has been added to Table 2 (formerly Table 1) titled: pathology, which showed the PTMs that are associated with particular cardiomyopathy. We hope this highlights the associations with Hsp90 PTMs that will ease the reviewers' concern. Unfortunately, current findings on the effect of Hsp90 PTMs on cardiac diseases are limited. So we were only able to identify two cardiomyopathy-related PTMs. For the rest of the PTMs, we can only speculate their roles in cardiac diseases as no reports are confirming their actions.
A bit more discussion of known modulation of hsp90 and cardiac outcomes would be beneficial (ex PMIDs: 30582930 and 29203715). Also, a discussion of any known mutations for gain- or loss-of-function (if there are any) in hsp90 and the impact on the heart.
Response:
We sincerely thank the reviewer for the suggestion. A new section 2.2 named “Pathophysiological Significance in Cardiomyopathy” (lines 315-355) has been added to summarize the physiological aspects of Hsp90 inhibition and its effects on the aforementioned pathways from the second section. Table 1 has also been added which highlights literature showing Hsp90 involvement and effects on the pathway. The table has the headers Pathway, Hsp90 Effect on Pathway, Cardiac Phenotype, and References. The “Pathway” column is the name of the pathway described in section 2. The “Hsp90 Effect on Pathway” section gives a 1-2 sentence description of the Hsp90-related findings of the paper. The “Cardiac Phenotype” section provides the pathological response associated with the findings from the paper. The “References” column provides the citation number so readers can easily find the information if they would like to.
In terms of Hsp90 mutations, we did not find literature that clearly connects Hsp90 mutations to cardiac diseases. More efforts on phylogenetic study will be needed in the future to expand our understanding in this area of research.